# Early Administration of Vancomycin Inhibits Pulmonary Embolism by Remodeling Gut Microbiota

**DOI:** 10.3390/jpm13030537

**Published:** 2023-03-17

**Authors:** Zhengyan Zhang, Huiling Chen, Jiating Huang, Shilong Zhang, Zhanming Li, Chaoyue Kong, Yuqin Mao, Bing Han

**Affiliations:** 1Center for Traditional Chinese Medicine and Gut Microbiota, Minhang Hospital, Fudan University, Shanghai 201199, China; 2Institute of Fudan-Minhang Academic Health System, Minhang Hospital, Fudan University, Shanghai 201199, China

**Keywords:** PE, gut microbiota, vancomycin, antibiotics, FMT

## Abstract

Pulmonary embolism (PE) is a common and potentially fatal condition in the emergency department, and early identification of modifiable risk factors for prevention and management is highly desirable. Although gut dysbiosis is associated with a high incidence of venous thromboembolism, the role and mechanisms of the gut microbiome in the pathogenesis of venous thromboembolism, especially PE, remain unexplored. Here, we attempted to elucidate the benefits of the gut microbiome in the pathogenesis of PE using multiple antibiotics and fecal microbiota transplantation (FMT) for early intervention in a classical mouse model of PE. The results showed that early administration of various antibiotics (except ampicillin) could inhibit pulmonary thrombosis to a certain extent and reduced mortality in young and old mice with PE. Among them, vancomycin has the best inhibitory effect on PE. With the help of gut microbiota sequencing analysis, we found that antibiotic treatment can reshape the gut microbiota; especially vancomycin can significantly improve the gut microbiota structure in PE mice. Furthermore, FMT could transfer vancomycin-modified gut microbes into mice and inhibit the pathogenesis of PE, possibly due to increased intestinal colonization by Parasutterella. These data elucidate the underlying molecular mechanism by which early administration of vancomycin can remodel the gut microbiota to suppress PE, providing new clues for clinical optimization and development of PE prevention strategies.

## 1. Introduction

Deep vein thrombosis and pulmonary embolism (PE) constitute venous thromboembolism [1]. Venous thromboembolism is a major global burden with approximately 10 million cases per year, therefore representing the third leading vascular disease after acute myocardial infarction and stroke [2]. Venous thromboembolism is a frequently recurring chronic disease that is associated with death, major bleeding associated with anticoagulation, and long-term disability. In the United States, the annual medical costs of venous thromboembolism are estimated at USD 7–10 billion [3]. Incidence is steadily rising due to an aging population and increased prevalence of comorbidities associated with venous thromboembolism, such as obesity, heart failure, and cancer, as well as increased sensitivity and widespread use of imaging tests to detect venous thromboembolism [4]. The average annual incidence rate increases exponentially with age, reaching as high as one case per hundred people over the age of 80 [5]. Beginning at age 45, the lifetime risk of developing venous thromboembolism is 8% [6]. In the International Cooperative Pulmonary Embolism Registry, the primary outcome—all-cause mortality rate at 3 months—associated with acute pulmonary embolism was 17%. The registry had no exclusion criteria and consecutively enrolled 2454 patients from 52 hospitals in seven countries in Europe and North America. PE was considered to be the cause of death in 45% of patients [7].

Sequelae of venous thromboembolism are also associated with severe disability and include post-thrombotic syndrome, which develops in 20–50% of patients with deep vein thrombosis, and chronic thromboembolic pulmonary hypertension, which complicates 0.1–4.0% of pulmonary embolisms [8]. Post-thrombotic syndrome can lead to chronic lower leg swelling, which can lead to trophic disorders including venous ulcers. Chronic thromboembolic pulmonary hypertension is defined as a mean pulmonary arterial pressure greater than 25 mm Hg for 6 months after the diagnosis of pulmonary embolism. It causes dyspnea and even sudden cardiac death with rest and exertion [9]. Deaths from PE usually occur within weeks after the diagnosis is made. The short-term mortality rate of PE varies widely and ranges from less than 2% in many patients with non-massive PE to more than 95% in patients who experience cardiorespiratory arrest [10,11]. Hence, given the associated morbidity and mortality, there is a strong need to identify modifiable risk factors to prevent thrombotic events and, by extension, adverse long-term outcomes.

The development of clinical thrombosis could be attributed to a combination of vessel wall damage, altered blood flow, and abnormal composition of the blood [12,13]. There is growing evidence that inflammation is an important risk factor for PE. Inflammation activates endothelial cells, platelets, and leukocytes to initiate coagulation. Activated leukocytes are a major source of procoagulant tissue factor-positive particles that may stimulate thrombus formation and growth. Neutrophil extracellular traps (NETs) composed of DNA, histones, and antimicrobial proteins provide erythrocytes, platelets, and procoagulant molecules with the ability to promote thrombosis [14]. Activation of the coagulation cascade can trigger the immune system, and inflammation is further associated with dysbiosis, increased intestinal permeability, and production of specific metabolites. With the development and progress of artificial intelligence, the combination of machine learning and electronic medical record (EHR) may reduce the risk of adverse outcomes by identifying previously unknown interventions. Arghya Datta et al. designed a machine learning model to determine the factors affecting the risk of hospital-acquired venous thromboembolism (HA-VTE) [15]. In addition to showing drug-drug interactions, machine learning sometimes also helps to remodel gut microbiome [16].

The gut microbiome plays a critical role in various inflammatory conditions (ranging from obesity and cardiovascular disease to autoimmune phenomena), and its modulation is a potential treatment option for these conditions [17,18,19]. Perturbations of the gut microbiome from various environmental or genetic factors can lead to activation of inflammatory pathways in vascular endothelial cells, platelets, and innate immune cells resulting in release of various coagulation proteins leading to a prothrombotic state [20,21,22]. The role of the gut microbiome in the pathogenesis of thromboembolism has not been fully elucidated. Many treatment options to modulate gut dysbiosis, such as fecal microbial transplantation (FMT), probiotics, and selective antibiotics, are currently being explored, but reports on efficacy are mixed [23]. In order to clarify the effect and mechanism of gut microbiota in the treatment of PE, here we used two models of broad-spectrum antibiotics and FMT treatment to study and found that vancomycin could significantly alter the gut dysbiosis in PE mice, reduced pathogen abundance, enhanced beneficial symbiotic anaerobic bacteria, and significantly reduced mortality in mice with PE. This study clarified the effect of broad-spectrum antibiotics in the treatment of PE, clarified the potential mechanism of vancomycin in the treatment of PE, and provided new clues for clinical optimization and development of prevention and treatment strategies for PE.

## 2. Materials and Methods

Mice. Male C57BL/6 mice (3 months and 14 months old) were purchased from Charles River Laboratory (Hangzhou, China) and maintained under pathogen-free (SPF) conditions in a temperature-controlled colony chamber (light/dark cycle for 12 h). Mice were fed the rodent chow and water ad libitum. After a week of adaptation, mice were used for the study. The animal study protocol was approved by the Ethics Committee of the Department of Experimental Animal Science, Fudan University.

Antibiotic treatments. For assessing the treatment potential of antibiotics, mice were divided into six groups: Saline, Ampicillin (200 mg/kg), Vancomycin (100 mg/kg), Metronidazole (200 mg/kg), Neomycin (200 mg/kg), and AVMN ((ampicillin (200 mg/kg), vancomycin (100 mg/kg), metronidazole (200 mg/kg), and neomycin (200 mg/kg)). The antibiotics were dissolved in saline, and then each group of mice was given the corresponding antibiotics dissolved in saline by gavage for 10 days (200 uL/mice).

Pulmonary Embolism Mouse Model. Mice were anesthetized with isoflurane. Then, the left side of the internal jugular vein was completely exposed by cutting the skin of the neck, and 15 IU/30 g thrombin was injected into the jugular vein. Then, the mortality rate was recorded in all groups within 30 min of the injection (the time of respiratory arrest lasted for at least 2 min). Dead mice caused by massive bleeding events were discarded. The left lung lobe was then removed, fixed in 10% formalin, and embedded in paraffin. Sections were stained with hematoxylin–eosin. Under the microscope, three fields (X10 objective, X10 ocular) were chosen at random in each section (one section per mouse). The thrombus area of lung tissue was evaluated in each field which was performed blind to groups [24,25].

Microbial DNA extraction and 16S rRNA gene sequencing. After the treatment of antibiotics and FMT, mouse fecal samples were collected and stored at −80 °C until processing. The genomic DNA of feces was extracted by DNA Extraction Kit (TIANGEN, Beijing, China). The Microbial 16S rDNA V3–V4 region was amplified by PCR. According to the previously published protocols and primers, the PCR amplification was performed in triplicate using the barcoded universal bacterial primers in a Gene Amp PCR-System 9700 (Applied Biosystems, Foster City, CA, USA). The PCR amplification products were sequenced using an Illumina HiSeq platform (Illumina MiSeq, SanDiego, CA, USA) [26].

Microbiome bioinformatic analysis. Gut microbiota α- and β-diversity analysis utilized QIIME and R. Differential abundance at the genus level was identified using R package DESeq2. To further analyze differentially abundant taxa responsible for the classification between two groups, an unsupervised RandomForest classification analysis was performed with the R package randomForest using 1000 trees as well as default settings. Metagenome functional content prediction was performed using PICRUSt (Version 1.1.1) [27]. LEfSe was used for linear discriminant analysis [28].

Fecal microbiota transplantation. For the transplant material preparation, feces were collected from the donor mice. Fecal samples weighing 80–100 mg (3–5 fresh feces pellets) were homogenized in 5 mL of PBS and then centrifuged at 8000 rpm (4 °C). The supernatant was discarded. This process was repeated 3 times to remove the impact of residual antibiotics in the bacterial suspension from donors [29,30]. Lastly, 1 mL of PBS was added to resuspend. The resulting homogenate was filtered by the 100 μm filter and used as the transplant material. Before FMT, recipient mice were treated for five consecutive days with 200 μL of an antibiotic cocktail by oral gavage to remove their own flora. The antibiotic cocktail contained ampicillin (200 mg/kg), vancomycin (100 mg/kg), metronidazole (200 mg/kg), and neomycin (200 mg/kg). Thereafter, recipient mice were given 200 μL of the fresh microbiota suspension by oral gavage three times a week for two weeks. All animal experiments were approved by the Ethical Committee of Minhang hospital, Fudan University.

Statistics. Data analyses were performed with either GraphPad Prism 8 (GraphPad, San Diego, CA, USA) or R (http://www.R-project.org/, accessed on 3 May 2019). Data were expressed as the mean ± standard error of mean (SEM). Survival difference was assessed by Kaplan–Meier survival curves. Unpaired two-tailed Student’s *t*-test and Log-rank tests were performed as indicated. *p* < 0.05 was considered to indicate a statistically significant difference.

## 3. Results

Vancomycin treatment reduces mortality in mice with pulmonary embolism. In order to explore the effect and mechanism of gut microbiota in the treatment of pulmonary embolism (PE), we treated mice by gavage with broad-spectrum antibiotics commonly used in clinical treatment to remodel the microbiota and then established an acute pulmonary embolism (APE) mouse model to observe the effect of different treatments on APE. First, three-month-old male mice were weighed and randomly divided into six groups, including control (NC; Sterile saline) (n = 8), vancomycin (Van; 100 mg/kg) (n = 8), ampicillin (Amp; 200 mg/kg) (n = 8), metronidazole (Met;200 mg/kg) (n = 8), neomycin (Neo; 200 mg/kg) (n = 8)), and the antibiotic cocktail group (AVMN; (ampicillin (200 mg/kg), vancomycin (100 mg/kg), metronidazole (200 mg/kg), and neomycin (200 mg/kg)) (n = 9). To remodel the gut microbiota in vivo, mice were continuously treated with the corresponding antibiotics by gavage for ten days (200 uL/mice). The body weight of the mice was examined to be unchanged before and after antibiotic treatment, thus ruling out the toxic effects of antibiotics on the mice. Mice were then injected with thrombin (15 IU/30 g) intravenously to induce APE. The results showed that within 5 min after thrombin injection, all the mice in the NC group died. Compared with the NC group, other antibiotic treatment groups except the ampicillin group had inhibitory effects on the death of mice caused by APE. Among them, vancomycin treatment can significantly reduce the mortality of APE mice, can prolong the survival of mice (40% within 5 min, 50% within 30 min), and has the best inhibitory effect on mortality caused by APE (Figure 1A,B). Next, we sacrificed the mice 30 min after thrombin injection and performed H&E staining on the lung tissue of the mice to observe the effect of antibiotic treatment on pulmonary thrombus in each group. Consistent with the mortality results, except for the NC and ampicillin-treated mice, which had extensive pulmonary thrombus areas, the other antibiotic-treated groups had reduced pulmonary thrombus; especially the vancomycin-treated mice had the least pulmonary thrombus area (Figure 1C,D). Since the incidence of pulmonary embolism in the population increases with age, we further observed the effect of different antibiotic treatments on pulmonary embolism in 14-month-old older mice (equivalent to 45 years in humans). The treatment scheme of antibiotics is the same as that of three-month-old mice. The results showed that, similar to the human population, acute pulmonary embolism developed more rapidly and more severely in older mice than in younger mice. Within 5 min after thrombin injection (15 IU/30 g), all mice in the NC group died (NC, n = 5); vancomycin could significantly reduce the mortality of APE mice to 40% (Van, n = 5), while other antibiotic-treated groups had a slight protective effect on APE mice (the lethality was 80%), and this phenotype remained unchanged until 30 min (Neo, n = 5;Amp, n = 5; Met, n = 5; AVMN, n = 10) (Figure 1E,F). Correspondingly, the area of thrombus formation in the lung tissue of the antibiotic-treated mice was less than that of the NC group, and the pulmonary thrombus in the vancomycin-treated group was the least (Figure 1G,H).

Antibiotic treatment reshapes gut microbiota in mice. The results in Figure 1 suggest that different antibiotic treatments can reduce pulmonary thrombosis and mortality in APE mice, most likely by altering the gut microbiota in mice. To assess how these antibiotics affect gut microbial community structure, we analyzed the alpha and beta diversity of gut microbiota across groups and compared microbial diversity within and between communities. First, fresh feces of three-month-old mice after antibiotic treatment were collected (NC, n = 6; Van, n = 6; Amp, n = 7; Neo, n = 5; Met, n = 5; AVMN, n = 5); bacterial DNA was extracted; and 16S rRNA gene sequencing was used for classification and analysis. The alpha diversity of the gut microbiota of mice in each group was analyzed and compared by calculating observed species (Figure 2A), Chao1 richness (Figure 2B), and Shannon diversity (Figure 2C). The results suggested that different antibiotic treatments resulted in different diversity and richness of gut microbiota in mice. Next, using the Bray–Curtis dissimilarity to identify differential clusters in principal coordinate analysis (PCoA), the beta diversity of the gut microbiota was assessed, and it was found that the clusters of gut microbiota were clearly separated across groups, and most importantly, vancomycin-treated group was significantly different from the other groups (Figure 2D). To further investigate the similarity between different samples, we constructed a hierarchical clustering tree at the OTU level using the unweighted UniFrac distances (left) and the component proportion of the bacterial phylum in each group (right), in which mice treated with the same antibiotic were grouped together, and the distances were also closest (Figure 2E). Our findings indicate that antibiotic treatment changes the structure of the gut microbial community in mice. The use of antibiotics can reshape the gut microbiome of mice due to their unique antimicrobial spectrum.

Vancomycin treatment improves the taxonomic composition of the gut microbiota in mice. To further identify candidate effector microorganisms that may contribute to the differences in APE mortality across groups, we next focused on differences in gut microbiota composition between the vancomycin-treated group and each of the other antibiotic-treated groups. To this end, we first compared enrichment of OTUs among all groups, which revealed enrichment for specific bacterial communities in each group at the phylum level and genus level (Figure 3A,B). To further investigate these findings, we conducted high dimensional class comparisons using linear discriminant analysis of effect size (LEfSe) that detected marked differences in the predominance of bacterial communities among all groups (Figure 3C,D): The Van group exhibited an enrichment of Deltaproteobacteria (class level). The NC group was dominated by Verrucomicrobia (class level). Clostridia and Bacteroidia (class level) were significantly enriched in the Amp group. The Met group had a predominance of Bacilli; the Neo was dominated by Erysipelotrichia (class level); and AVMN had an enrichment of Gammaproteobacteria at class level (Figure 3C,D). We then explored whether the gut microbiome could be separated according to the comparative heatmap of OTU abundance at genus level (Figure 3E). Logistic regression and LASSO were used to screen the genus features. Differential segregation of different taxonomic communities can be seen according to antibiotic treatment of APE mice. Compared with the other five groups, the Van group showed enrichment in Parasutterella, Bilophila, Enterobacter, Proteus, Providencia, and Morganella (genus level), while the relative abundance of Roseburia, Bacteroides, and Muribaculum (genus level) was significantly lower (Figure 3E).

Vancomycin-induced gut microbiota changes protect against pulmonary embolism. Based on the analysis of the microbial communities in each group after antibiotic treatment in Figure 3, we next focused on the nine microbial communities that were more variable in the vancomycin-treated group compared to the other groups at the genus level. Through further verification, it was found that the six genera of Parasutterella, Bilophila, Enterobacter, Proteus, Providencia, and Morganella in the vancomycin treatment group were more abundant (Figure 4A), while the other three genera Bacteroides, Muribaculum, and Roseburia were less than in the NC group (Figure 4B). To further explore the relationship between gut microbiota and pulmonary embolism, we focused on the association of five bacterial genera significantly altered in the vancomycin group with prognosis in APE mice. After the establishment of the APE model, the relative abundance (genus level) of Parasutterella, Bilophila, and Enterobacter in surviving mice (n = 7) after different antibiotic treatments was slightly higher than those in dead mice (n = 27), whereas the levels of Bacteroides and Muribaculum were slightly lower in surviving mice than in dead mice (Figure 4C). We classified APE mice into high and low categories based on the average relative abundance of these five bacterial genera (Parasutterella, Bilophila, Enterobacter, Bacteroides, and Muribaculum) in surviving mice following antibiotic treatment of APE. Kaplan–Meier was used to analyze the correlation between the relative abundance of these five bacterial genera and the survival of APE mice by log-rank test. We found that APE mice with high relative abundance of Parasutterella had higher survival rates than mice with low Parasutterella abundance (high, n = 7; low, n = 27; *p* = 0.046). APE mice with higher relative abundance of Enterobacter had better prognosis (high, n = 10; low, n = 24; *p* = 0.031). In addition, APE mice with high Bilophila content had a slightly higher survival rate (high, n = 9; low, n = 25; *p* = 0.385). Conversely, APE mice with lower relative abundances of Bacteroides or Muribaculum had better prognosis (Bacteroides, high, n = 20; low, n = 14; *p* = 0.042; Muribaculum, high, n = 19; low, n = 15; *p* = 0.016) (Figure 4D).

Transfer of vancomycin-improved gut microbiota to mice via FMT inhibits PE. The above results highly suggest that vancomycin treatment can prevent pulmonary embolism by modulating the gut microbiota in mice. Next, we used the fecal microbial transplantation (FMT) method to further verify this conclusion. FMT is the transplantation of a donor’s fecal samples by either oral or rectal delivery to restore gut microbiota homeostasis. Currently, the role of FMT in metabolic syndrome has been studied in animal models and humans with favorable results [23,31]. FMT has been reported to be a safe and effective treatment option for recurrent C. difcile infection with remission rates of over 80% compared to antibiotics alone [32,33]. Before FMT, recipient mice were treated for five consecutive days with 200 μL of an antibiotic cocktail by oral gavage to remove their own flora as “receptors” mice. The antibiotic cocktail contained ampicillin (200 mg/kg), vancomycin (100 mg/kg), metronidazole (200 mg/kg), and neomycin (200 mg/kg). The feces of the mice in the NC group and the vancomycin-treated group were collected as donors, and the fecal microorganisms were transplanted into the “microbiota-deficient mice” (receptors). Thereafter, recipient mice were given 200 μL of the fresh microbiota suspension by oral gavage three times a week for two weeks. At the same time, the mice were gavaged with normal saline as a control group. Then, APE modeling was performed, and the specific experimental design was shown in Figure 5A. The recipient mice were checked for no change in body weight before and after FMT, thereby excluding the toxic side effects of FMT in mice. Thrombin was administered intravenously to recipient mice to induce APE. Within 5 min, both groups of recipient mice that were gavaged with normal saline (Saline group, n = 10) and feces in the NC group (R-NC group, n = 7) developed APE, resulting in massive death (mortality > 80%). However, the recipient mice transplanted with vancomycin-treated feces (R-Van group, n = 11) had significantly reduced APE-induced death (50% mortality) (Figure 5C), and survival rate was also prolonged to some extent (Figure 5D). Consistently, mice were sacrificed 30 min after thrombin injection, and H&E staining of mouse lung tissue showed that the R-Van group had less pulmonary thrombus, and the area of pulmonary thrombus was significantly smaller than the other two groups (Figure 5E,F). These results suggest that transfer of vancomycin-improved gut microbiota to mice via FMT also inhibits pulmonary embolism, similar to vancomycin treatment alone.

Vancomycin may inhibit pulmonary embolism by increasing Parasutterella. Taken together, since FMT was used to transfer vancomycin-improved gut microbiota to recipient mice, and vancomycin alone was used, both models inhibited mortality in mice with pulmonary embolism. Then, these two models are likely to regulate similar gut microbiota and exert their efficacy through the same mechanism. Therefore, we further analyzed the gut microbiota of FMT-treated recipient mice to try to find the key strains that vancomycin inhibits PE. Using the Bray–Curtis dissimilarity to identify differential clusters in principal coordinates analysis (PCoA) to assess beta diversity of gut microbiota, the results are consistent with Figure 2D, with clear separation of animal clusters between R-NC and R-Van groups. Next, high-dimensional class comparisons were performed using linear discriminant analysis of effect size (LEfSe), which detected significant differences in bacterial community dominance among groups and enriched bacteria between the R-NC and R-Van groups. Communities were significantly different (Figure 6B). Further, Logistic regression and LASSO were used to screen genus features, and the gut microbiome was analyzed according to the genus-level OTU abundance comparison heatmap, which could see differential segregation of taxonomic communities between the R-NC and R-Van groups (Figure 6C). Based on our results above, treatment with vancomycin alone mainly caused five significant changes in bacterial genera that were strongly associated with mortality in APE mice (Figure 4). So, next we focus on the changes in these five bacteria. First, Enterobacter was not found in the differential microbial communities of the two groups of recipient mice, suggesting that this bacterial genus likely did not colonize the recipient mice efficiently by FMT. In addition, Bacteroides were slightly elevated in the R-Van group compared with the R-NC group, which was different from the changes in APE mice treated with vancomycin alone, indicating that Bacteroides are not the key bacteria for vancomycin-inhibited pulmonary embolism. Importantly, Parasutterella was significantly increased in the R-Van group compared with the R-NC group (Figure 6D), which is consistent with the above changes in APE mice treated with vancomycin alone. The relative abundance of these three bacterial genera was closely related to the survival rate of APE mice (Figure 6E). In particular, the relatively high relative abundance of Parasutterella in the recipient mice of the R-Van group explained the high survival rate of APE mice, suggesting that Parasutterella may be the key bacteria for vancomycin to inhibit PE.

## 4. Discussion

Pulmonary embolism (PE) is a common and potentially lethal condition in the emergency department requiring early and accurate management, and given its high morbidity and mortality, identification of modifiable risk factors to prevent PE is highly desirable. The pathogenesis of PE is complex and occurs from the additive effects of genetic and environmental risk factors [34], of which inflammation is an important risk factor for PE. Inflammation not only initiates coagulation, but also leads to consumptive coagulopathy and increases in proinflammatory cytokines, chemokines, and various leukocyte subtypes [35,36]. Inflammatory states of several diseases, such as obesity, sepsis/infection, inflammatory bowel disease (IBD), and intestinal failure (IF), have been reported to be associated with a high incidence of intestinal dysbiosis and venous thromboembolism [37,38,39,40,41,42]. However, the molecular mechanisms of the gut microbiome in the pathogenesis of venous thromboembolism, especially PE, remain largely unexplored.

Using a mouse model of PE, we performed early intervention using multiple broad-spectrum antibiotics and FMT for the first time to clarify the benefits of the gut microbiome in the pathogenesis of PE. The results show that the early administration of various antibiotics (except ampicillin) in young and old mice can inhibit pulmonary thrombosis to a certain extent, reduce the mortality rate of PE, and prolong the survival time of mice. Among them, vancomycin has the best effect on inhibiting the pathogenesis of pulmonary embolism. With the help of gut microbiota sequencing analysis, we found that antibiotic treatment can reshape the intestinal microbiota of mice; especially vancomycin can significantly improve the intestinal microbiota structure of mice with PE. Further studies found that FMT could transfer vancomycin-improved gut microbes into mice to exert anti-pulmonary embolism effects, possibly due to increased intestinal colonization of Parasutterella. These data demonstrate the importance of early and timely use of vancomycin to inhibit the pathogenesis of pulmonary embolism, elucidate the potential molecular mechanism of vancomycin inhibiting pulmonary embolism by remodeling gut microbes, and also provide new clues for clinical optimization and development of PE prevention strategies.

Important risk factors of thrombosis include cancer, surgery, inflammation, bed restraint, major trauma, long journeys, pregnancy, oral contraceptives, previous venous thromboembolism, and bacterial infections [43]. Cancer is an independent and major risk factor for venous thromboembolism (VTE) [44], including deep vein thrombosis (DVT) and pulmonary embolism (PE). Of all first VTE events, 20% to 30% are malignancy-associated, and VTE is the second leading cause of death in patients with malignancy [45]. The main reason is that measures such as direct tumor invasion, radiotherapy and chemotherapy, or central venous catheterization will directly damage the blood vessel wall to activate the coagulation system, resulting in an abnormal increase in platelets, placing the body in a hypercoagulable state and abnormal function of the fibrinolytic system, thereby promoting thrombosis [46,47]. Once VTE occurs in cancer patients, the difficulty of treatment increases; the survival period is shortened; and the consequences are serious. Therefore, early diagnosis and prevention are particularly important. In the process of cancer treatment, patients are susceptible to infection by pathogenic microorganisms due to low immunity and organ failure. It is generally prevented and treated with antibiotics. It is well known that the role of the gut microbiome in various aspects of human health is increasingly recognized as having a major impact on host physiology. The human body is colonized by trillions of resident microorganisms consisting of a large number of commensal obligate anaerobic bacteria and potentially pathogenic bacteria [48]. Dysbiosis is defined as a microbial imbalance that typically manifests as a decrease in microbial diversity: for example, a decrease in commensal anaerobic gut bacteria and an overgrowth of pathobionts such as the bacterial family Enterobacteriaceae (ENTERO) [49]. Antibiotics can eliminate pathogenic bacteria as well as beneficial bacteria, which can lead to microbial imbalances affecting local and systemic pathophysiological processes. In this paper, we explored for the first time the efficacy of early use of various commonly used antibiotics to inhibit pulmonary embolism, which may provide new clues for the use of antibiotics in clinical cancer patients and the treatment of pulmonary embolism. There are still many unidentified factors that need to be further verified in the future, such as the timing of antibiotic use, the impact of baseline gut microbiota on efficacy, and the possibility of combined FMT therapy.

Here, either vancomycin alone or FMT to transfer vancomycin-improved gut microbiota into recipient mice significantly suppressed mortality in mice with PE. Prescribing vancomycin or FMT in patients at VTE risk for its prevention seems impossible in a near future. We found that the flora had an impact on the blood coagulation function of mice through FMT experiment. Transferring vancomycin-treated gut microbiota into recipient mice significantly suppressed mortality in mice with PE. Are there certain flora that are beneficial to the prevention of patients at VTE risk? Further studies found that the gut microbiota analysis of both models suggested that vancomycin may inhibit PE mainly by regulating the colonization of Parasutterella in the gut and increasing its abundance. Parasutterella has been recognized for about 10 years as a genus of Betaproteobacteria, which has been defined as a member of the healthy fecal core microbiome in the human gastrointestinal tract [50]. Based on sequences reported in the Ribosomal Database Project (RDP), members of the genus Parasutterella have also been found in a variety of host species, including mouse, rat, dog, pig, chicken, turkey, and calf [51]. Interestingly, Parasutterella was recently reported to use succinic acid as a fermentative end-product, and its production of succinic acid was even greater than that of Bacteroides fragilis, one of the well-identified succinate producers [52]. Succinate, as one of the key intermediate metabolites produced by gut microbiota, plays an important role in cross-feeding metabolic pathways [53]. Studies have shown that Parasutterella is transmitted between mothers and vaginally born infants, with a gradual increase in relative abundance up to 12 months of age, suggesting that Parasutterella is one of the early colonizers in the neonatal gut and increases in response to dietary change and host development [54]. Using a germ-free mouse model study of neonatal microbiota reconstitution, bacterial-derived succinate was found to promote the colonization of strict anaerobic bacteria, Clostridiales, to protect mice from infection [55]. As one of the early colonizers, as well as a succinate-producing symbiotic bacteria, Parasutterella may play a role in microbial interactions and infection resistance, especially early in life. Furthermore, there is increasing evidence that the relative abundance of Parasutterella is associated with different host health outcomes, such as inflammatory bowel disease, obesity, diabetes, and fatty liver disease. An inverse correlation between Parasutterella abundance and high-fat diet (HFD)-induced metabolic phenotypes, including hypothalamic inflammation, has been observed in multiple animal models and human studies [56,57,58]. Furthermore, patients with Clostridium difficile infection (CDI) showed a significant increase in the abundance of Proteobacteria in the gut; however, within the phylum of Proteobacteria, Parasutterella in CDI patients and asymptomatic carriers was significantly lower than in healthy controls [59]. To our knowledge, our observations are the first to show that Parasutterella abundance increases with vancomycin administration and is inversely associated with the pulmonary embolism phenotype. Our study provides clues for expanding new biological effects of Parasutterella in inhibiting PE.

In conclusion, we demonstrated the effects of various antibiotics in a mouse model of PE for the first time and found that early administration of antibiotics could modulate the mouse gut microbiome, thereby inhibiting pulmonary thrombosis and reducing PE mortality to a certain extent. Among them, vancomycin has the best efficacy in inhibiting PE. These data highlight a key cross-talk between gut microbiota and pulmonary thrombosis during antibiotic treatment. Furthermore, with the aid of gut microbiota sequencing and FMT approaches, we found that vancomycin may inhibit PE by improving gut microbiome composition mainly by increasing the gut colonization of Parasutterella. These data elucidate the underlying molecular mechanism by which vancomycin inhibits the pathogenesis of PE and provide new clues for clinical optimization and development of PE prevention strategies.

## Figures and Tables

**Figure 1 jpm-13-00537-f001:**
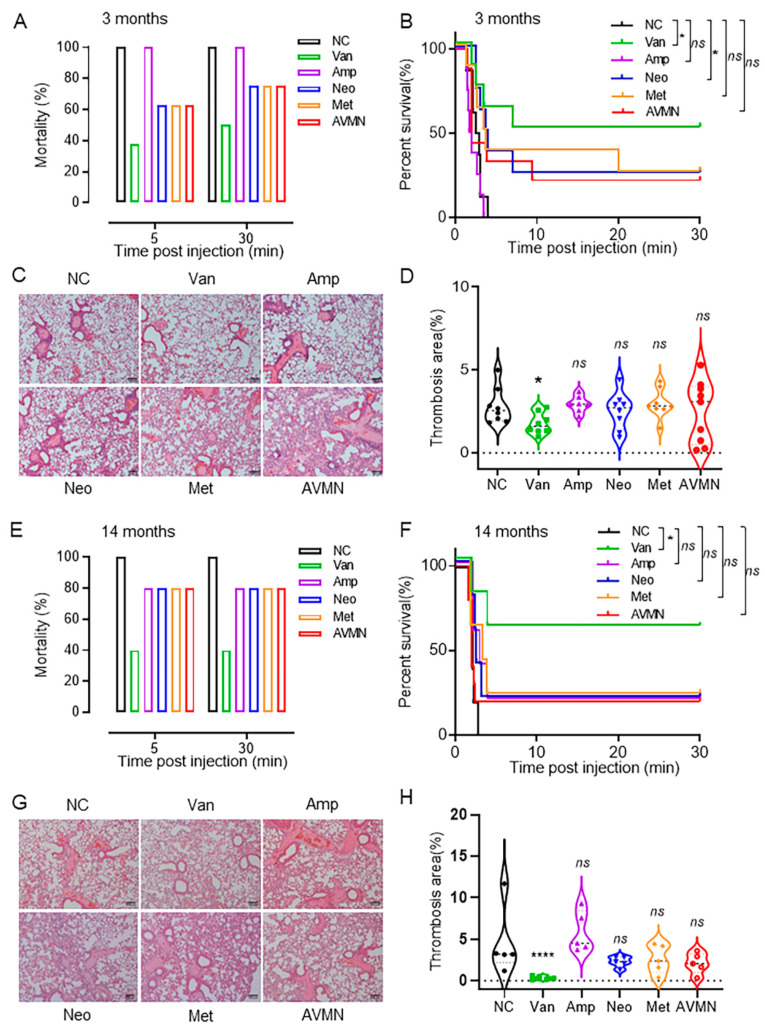
Vancomycin treatment reduces mortality in mice with pulmonary embolism. Three-month-old male mice were randomly divided into six groups including control (NC), vancomycin (Van) (n = 8), ampicillin (Amp) (n = 8), metronidazole (Met) (n = 8), neomycin (Neo) (n = 8), and an antibiotic cocktail Group (AVMN) (n = 9). (**A**) Mortality of PE mice in each group during the 30-min observation period. (**B**) Kaplan–Meier survival percentage of the mice in each group. (**C**) Representative images of H&E staining of lung tissue sections obtained from each group of mice, scale bar: 100 µm. (**D**) Quantitative analysis of thrombus area in lung tissue. (**E**–**H**) A model of pulmonary embolism was induced using 14-month-old male mice. NC, n = 5; Van, n = 5; Neo, n = 5; Amp, n = 5; Met, n = 5; AVMN, n = 10. Mortality (**E**) and Kaplan–Meier survival percentage (**F**) of PE mice in each group during the 30-min observation period. Representative images of mouse lung tissue sections of each group after H&E staining (**G**), scale bar: 100 µm. Quantitative analysis of thrombus area in lung tissue (**H**). ** p* < 0.05, ***** p* < 0.0001.

**Figure 2 jpm-13-00537-f002:**
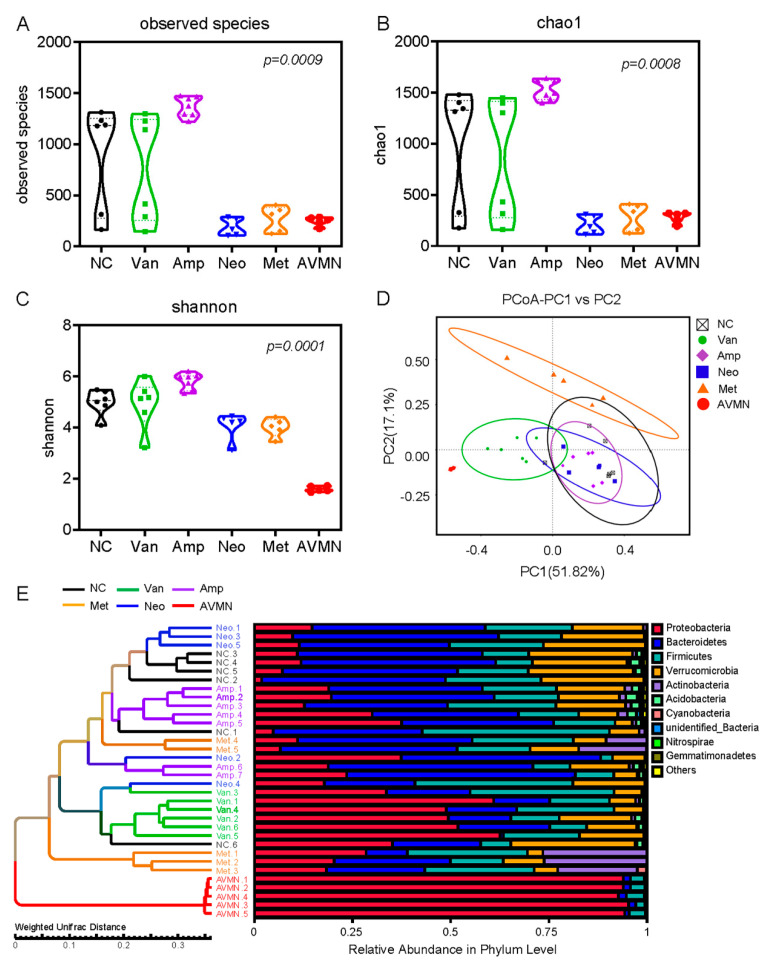
Antibiotic treatment reshapes gut microbiota in mice. (**A**–**C**) Comparison of alpha-diversity of gut microbiota among all groups of mice using observed—species (**A**), the Chao1 index (richness) (**B**) and Shannon’s index (diversity) (**C**). Analyses were performed using Kruskal—Wallis test. In the box plots, upper and lower hinges correspond to the first and third quartiles; the center line represents the median; and whiskers indicate the highest and lowest values. (**D**) Principal coordinate analysis (PCoA) using the Bray—Curtis dissimilarity metric of beta-diversity among samples of the six groups of mice analyzed. Each dot represents an individual mouse. PCo1 and PCo2 represent the percentage of variance explained by each coordinate. (**E**) UPGMA tree on the OUT level representing the beta-diversity analysis of all groups analyzed using the weighted—UniFrad and the community map of dominant phyla in each group. NC, n = 6; Van, n = 6; Amp, n = 7; Neo, n = 5; Met, n = 5; AVMN, n = 5.

**Figure 3 jpm-13-00537-f003:**
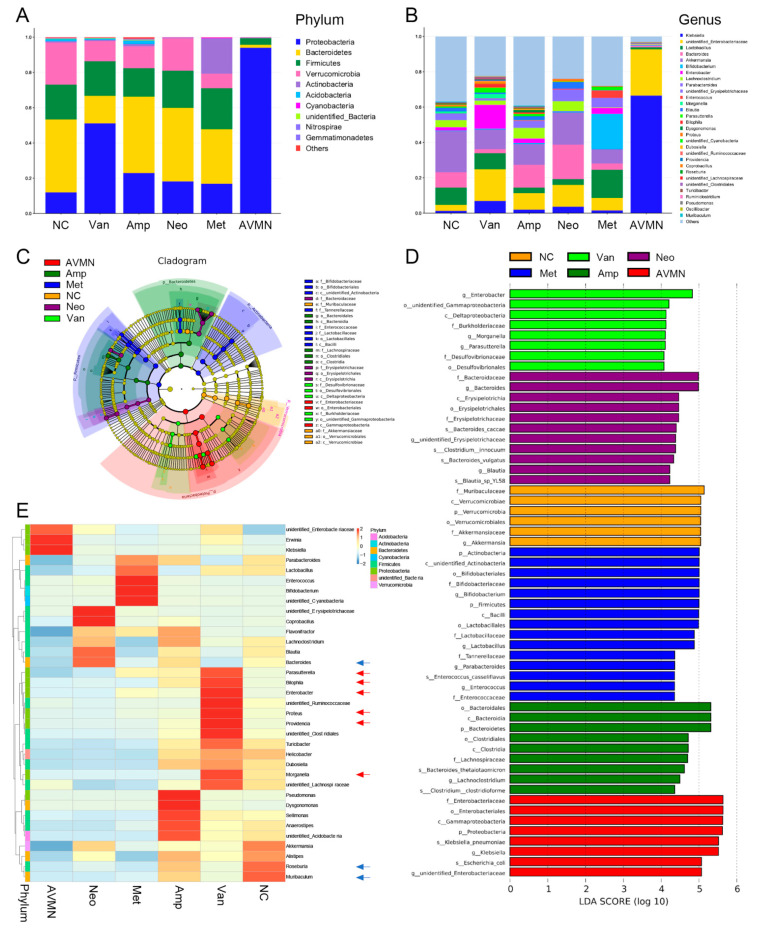
Vancomycin treatment improves the taxonomic composition of the gut microbiota in mice. (**A**,**B**) Bar plots displaying average relative abundance of prevalent microbiota at phylum level (**A**) and genus level (**B**) in the six groups studied. (**C**) Taxonomic cladogram obtained from LEfSe analysis showing bacterial taxa (family, class, order) that was differentially abundant among six groups of mice. Each node represents a specific taxonomic type. Each color represents mice treated with the same antibiotic. (**D**) Results of LEfSe analysis showing bacterial taxa that were significantly different in abundance among all groups. The criteria for feature selection are LDA score (log10) > 4. p, phylum; c, class; o, order; f, family; g, genus. (**E**) Heatmap of selected most differentially abundant features at the genus level in each group. The blue color represents less abundant; lighter yellow color represents intermediate abundance; and red represents the most abundant. The red arrow indicates the bacteria enriched, and the blue arrow indicates low abundance of bacteria in the Van group compared with the other five groups. NC, n = 6; Van, n = 6; Amp, n = 7; Neo, n = 5; Met, n = 5; AVMN, n = 5.

**Figure 4 jpm-13-00537-f004:**
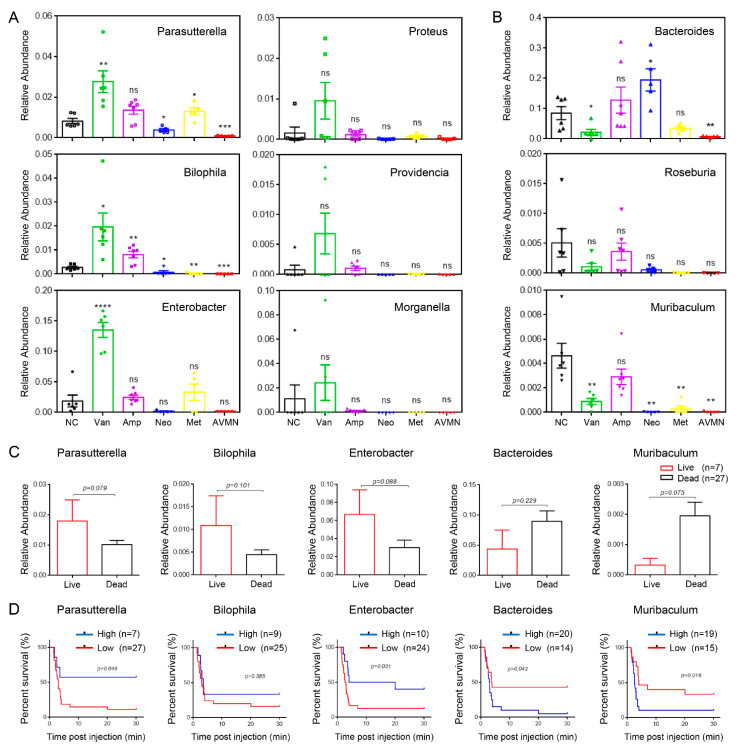
Vancomycin-induced gut microbiota changes protect against pulmonary embolism. (**A**,**B**) Bar chart showing the relative abundance of gut microbes enriched (**A**) and decreased (**B**) at genus level in vancomycin-treated mice, compared with NC group. Each dot represents a mouse. Van, Amp, Neo, Met, and AVMN group were compared with NC group. (**C**) Bar chart showing the relative abundance of gut microbes at genus level in live and dead mice with APE. (**E**) Kaplan–Meier estimates for survival probability of APE mice based on the abundance levels of microbes. * *p* < 0.05, ** *p* < 0.01, *** *p* < 0.001, **** *p* < 0.0001.

**Figure 5 jpm-13-00537-f005:**
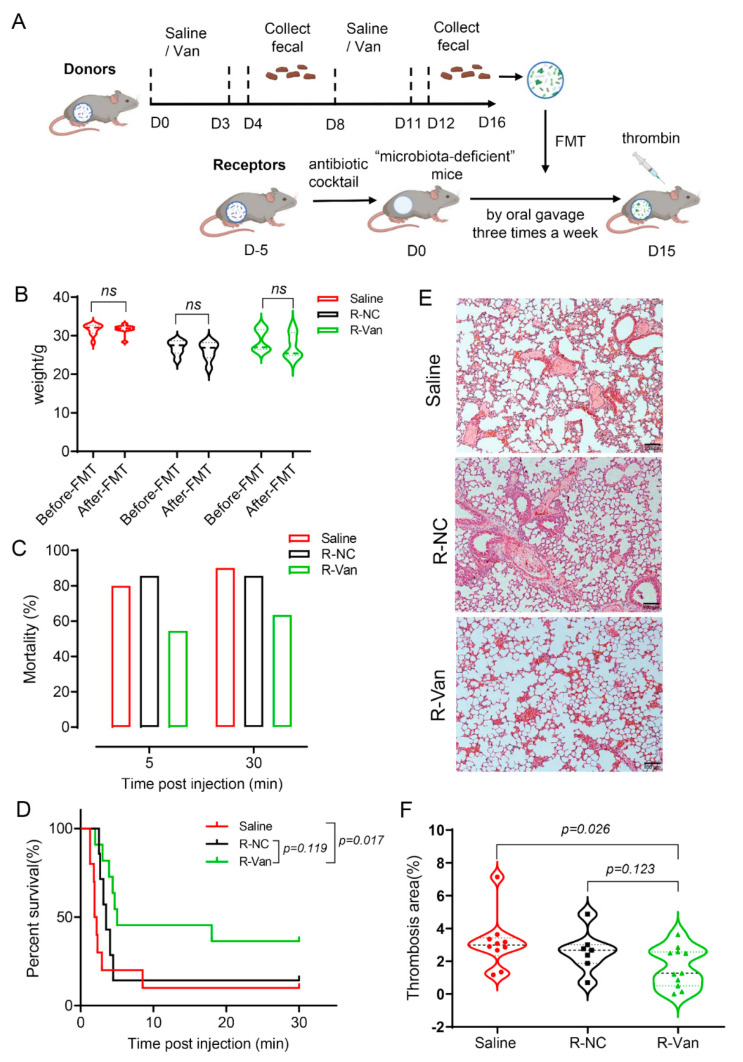
Transfer of vancomycin—improved gut microbiota to mice via FMT inhibits PE. (**A**) Experimental design of FMT: saline—treated mice and vancomycin-treated mice were used as donors to provide gut microbial. The AVMN—treated mice were receptors. (**B**) Weight of mice before and after FMT. (**C**) Number (percentage, %) of PE mice treated with the FMT and saline dead at 5 and 30 min post—thrombin injection. (**D**) Survival rate of APE mice treated with the FMT, and saline was recorded during a 30—minute observation period. The effect of FMT on survival was significant by log—rank test. R—Van vs. Saline, *p* = 0.0173; R—Van vs. R—NC, *p* = 0.119; *p* = 0.003. (**E**) H&E staining of lung sections from mice PE mice treated with the FMT and saline. Scale bar: 100 µm. (**F**) Quantitative analysis of the thrombus area in panel. These results represent the mean ± sem. Unpaired *t* test. Saline group, n = 10; R—NC group, n = 7; R—Van group, n = 11.

**Figure 6 jpm-13-00537-f006:**
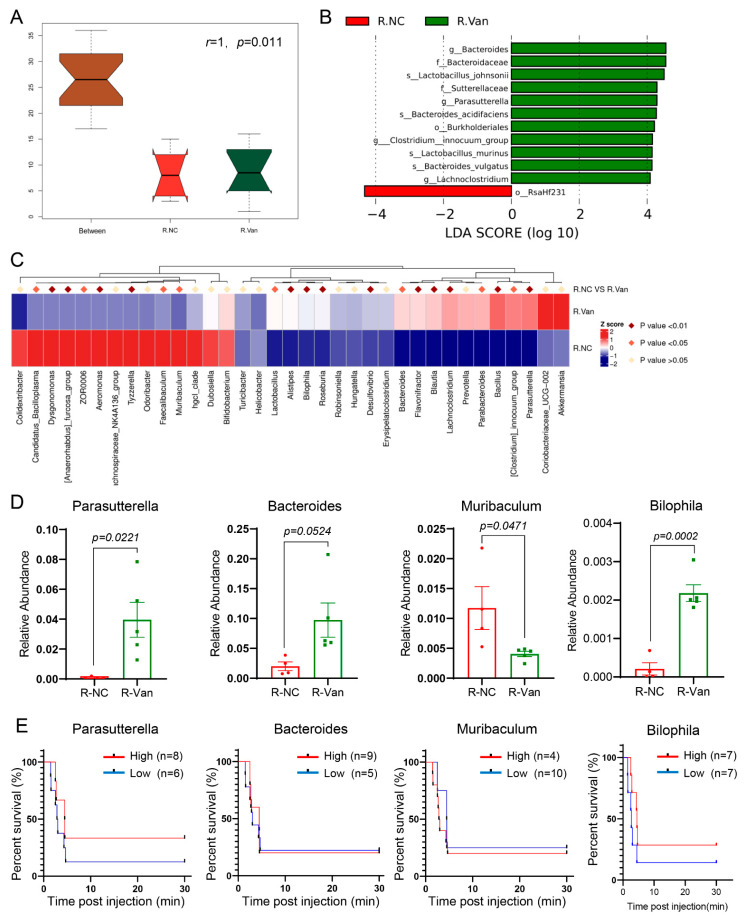
Vancomycin may inhibit pulmonary embolism by increasing Parasutterella. (**A**) Anosim of beta-diversity between R.NC and R.Van. (**B**) Results of LEfSe analysis showing bacterial taxa that were significantly different in abundance between R.NC and R.Van. The criteria for feature selection are LDA score (log10) > 4. p, phylum; c, class; o, order; f, family; g, genus. (**C**) Heatmap of selected most differentially abundant features at the genus level between R.NC and R.Van group. The blue color represents less abundant; white color represents intermediate abundance; and red represents the most abundant (**D**) Bar chart showing the relative abundance of gut microbes at genus level in R-NC and R-Van group. (**E**) Kaplan–Meier estimates for survival probability of APE mice based on the abundance levels of microbes. R-NC group, n = 4; R-Van group, n = 5.

## Data Availability

The datasets used and/or analysed during the current study are available from the corresponding author on reasonable request.

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
