# Peer review of "Early Administration of Vancomycin Inhibits Pulmonary Embolism by Remodeling Gut Microbiota"

_jpm, 2023, doi:10.3390/jpm13030537_

Round 1

Reviewer 1 Report

In this manuscript authors have reported the importance of administration of vancomycin can inhibit pulmonary embolism by remodelling gut microbiota. The paper is important as in today's world it has been reported that gut microbata remodelling can be advantageous in preventing and treatment of many diseases. However in this world of Artificial intelligence some inputs should be given in this paper as it would help to enrich the contant of the manuscript.

In the introduction authors should mention that how machine learning has shown that it can reduce hospital risk of thromboembolism using electronic data  and should mention this important paper (A. Datta et al., "‘Black Box’ to ‘Conversational’ Machine Learning: Ondansetron Reduces Risk of Hospital-Acquired Venous Thromboembolism," in IEEE Journal of Biomedical and Health Informatics, vol. 25, no. 6, pp. 2204-2214, June 2021, doi: 10.1109/JBHI.2020.3033405.)

Secondly authors should also mention that although Machine learning has shown NSAIDS cause drug drug interactions which can be fatal but sometime it can also help in remodelling gut microbiome and should cite this important paper as well (Datta A, Flynn NR, Barnette DA, Woeltje KF, Miller GP, Swamidass SJ (2021) Machine learning liver-injuring drug interactions with non-steroidal anti-inflammatory drugs (NSAIDs) from a retrospective electronic health record (EHR) cohort. PLoS Comput Biol 17(7): e1009053. https://doi.org/10.1371/journal.pcbi.1009053)

Author Response

Dear reviewer,

We appreciate your serious efforts on our manuscripts entitled “Early administration of vancomycin inhibits pulmonary embolism by remodeling gut microbiota (Manuscript ID jpm-2203078)". According to your comments, we have carefully revised it. We believe that the quality of this revised version has been further improved. We are looking forward that you find that it is acceptable for publication in JPM. The point-to-point responses are as following:

Response:  

Thanks for your constructive comments. Firstly, we have further improved the English writing of our manuscript. And these references have been added to this revised manuscript (reference 15 and 16), which are very interesting and valuable to the introduction for our study.

“With the development and progress of artificial intelligence, the combination of machine learning and electronic medical record (EHR) may reduce the risk of adverse outcomes by identifying previously unknown interventions. Arghya Datta et al. designed a machine learning model to determine the factors affecting the risk of hospital-acquired venous thromboembolism (HA-VTE). In addition to showing drug-drug interactions, machine learning sometimes also helps to remodel gut microbiome.” We have added these contents to the introduction (reference 15 and 16) (Page20, Line 79-85).

Reviewer 2 Report

This is potentially an important study. However, methodology of the research is poorly described. It has to be improved. Every step of the study must be described in details, including number of animals, timing of drug administration, doses of antibiotics, details of all interventions, etc.

Author Response

Dear reviewer,

We appreciate your serious efforts on our manuscripts entitled “Early administration of vancomycin inhibits pulmonary embolism by remodeling gut microbiota (Manuscript ID jpm-2203078)". According to your comments, we have carefully revised it. We believe that the quality of this revised version has been further improved. We are looking forward that you find that it is acceptable for publication in JPM. The point-to-point responses are as following:

Response:

We have carefully revised methodology of the research in “Result” and “Materials and methods”.

“First, 3-month-old male mice were weighed and randomly divided into six groups, including control (NC; Sterile saline)(n=8), vancomycin (Van; 100 mg/kg) (n=8), ampicillin (Amp; 200 mg/kg) (n=8), metronidazole (Met;200 mg/kg) (n=8), neomycin (Neo; 200 mg/kg) (n=8) ), and the antibiotic cocktail group (AVMN; (ampicillin (200 mg/kg), vancomycin (100 mg/kg), metronidazole (200 mg/kg) and neomycin (200 mg/kg)) (n=9). To remodel the gut microbiota in vivo, mice were continuously treated with the corresponding antibiotics by gavage for ten days (200ul/mice).” (Page 6, Line 169-176)

“The treatment scheme of antibiotics is the same as that of 3-month-old mice. The results showed that, similar to the human population, acute pulmonary embolism developed more rapidly and more severely in older mice than in younger mice. Within 5 minutes after thrombin injection (15 IU/30 g), all mice in the NC group died(NC, n=5), vancomycin could significantly reduce the mortality of APE mice to 40%(Van, n=5), while other antibiotic-treated groups had a slight protective effect on APE mice (the lethality was 80%), and this phenotype remained unchanged until 30 minutes(Neo, n=5;Amp, n=5; Met, n=5; AVMN, n=10 ) (Figure 1E-F).” (Page 7, Line194-202)

“First, fresh feces of 3-month-old mice after antibiotic treatment were collected (NC, n=6; Van, n=6; Amp, n=7; Neo, n=5; Met, n=5; AVMN, n=5), bacterial DNA was extracted, and 16S rRNA gene sequencing was used for classification and analysis.” (Page 9, Line 226-229)

“Before FMT, recipient mice were treated for five consecutive days with 200μl of an antibiotic cocktail by oral gavage to remove their own flora as “receptors” mice. The antibiotic cocktail contained ampicillin (200 mg/kg), vancomycin (100 mg/kg), metronidazole (200 mg/kg) and neomycin (200 mg/kg). The feces of the mice in the NC group and the vancomycin-treated group were collected as donors, and the fecal microorganisms were transplanted into the “microbiota-deficient mice” (receptors). Thereafter, recipient mice were given 200μl of the fresh microbiota suspension by oral gavage three times a week for two weeks.” (Page 15, Line 345-352)

Reviewer 3 Report

Dear Authors!

Thank you so much for such an interesting study conducted. I believe two very important issues regarding suggested connection of gur microbiota correction and PE must be addressed.

1. Even if preventive effect of vancomycin and FMT does exist I can hardly imagine a clinical implication of this. To prescribe this medication in patients at VTE risk for its prevention seems impossible, at least in a near future. You have to clarify possible cohorts of patients in whom this prescription could be justified in the future. If you mean "obesity, sepsis/infection, inflammatory bowel disease (IBD), and intestinal failure", then you have to present data on a risk of VTE related to those conditions to support a need for seek additional preventive measures beyond anticoagulants which are quite effective. Your introduction and discussion have to include more practice oriented background.

2. As all the antibiotics that you used in the study have such a side effect as thrombocytopenia one can explain your findings simply by that. This side effect may lead to hypocoagulability. As different antibiotics can lead to thrombocytopenia to a different extent then it may explained different mortality rates. Without knowing what happened to coagulation in mice who received antibiotics to suggest that this is microbiota changes prevents PE is not correct. 
I believe that you have to discuss that there must be another explanation of your findings. 

Minor remarks

in Introduction

1.      Please, change the sentence to be as

“Post-thrombotic syndrome can lead to chronic lower leg swelling, which can lead to trophic disorders  including venous ulcers”

2.      In fact, Virchow did not propose a triad. It was given his name later. Please, correct this statement and use for this a more relevant reference.

Author Response

Dear reviewer,

We appreciate your serious efforts on our manuscripts entitled “Early administration of vancomycin inhibits pulmonary embolism by remodeling gut microbiota (Manuscript ID jpm-2203078)". According to your comments, we have carefully revised it. We believe that the quality of this revised version has been further improved. We are looking forward that you find that it is acceptable for publication in JPM. The point-to-point responses are as following:

1. Even if preventive effect of vancomycin and FMT does exist I can hardly imagine a clinical implication of this. To prescribe this medication in patients at VTE risk for its prevention seems impossible, at least in a near future. You have to clarify possible cohorts of patients in whom this prescription could be justified in the future. If you mean "obesity, sepsis/infection, inflammatory bowel disease (IBD), and intestinal failure", then you have to present data on a risk of VTE related to those conditions to support a need for seek additional preventive measures beyond anticoagulants which are quite effective. Your introduction and discussion have to include more practice oriented background.

Response:

Thank you for your advice. we agree with this view that, at least in the near future, more research data and clinical experimental demonstration are needed for the use of vancomycin and FMT in the prevention and treatment of patients at VTE risk. However, the purpose of this study is more to find beneficial bacteria that can prevent patients at risk of VTE and possible methods to improve the flora. In the future, develop prebiotics or microbiota capsules for potential beneficial bacteria. We added this part to the discussion.

“Prescribing vancomycin or FMT in patients at VTE risk for its prevention seems impossible in a near future. But we found that the flora had an impact on the blood coagulation function of mice through FMT experiment. Transferring vancomycin-treated gut microbiota into recipient mice significantly suppressed mortality in mice with PE. Is there certain flora that is beneficial to the prevention of patients at VTE risk? Further studies found that the gut microbiota analysis of both models suggested that vancomycin may inhibit PE mainly by regulating the colonization of Parasutterella in the gut and increasing its abundance.” (Page 23, Line 495 and 502).

2.As all the antibiotics that you used in the study have such a side effect as thrombocytopenia one can explain your findings simply by that. This side effect may lead to hypocoagulability. As different antibiotics can lead to thrombocytopenia to a different extent then it may explained different mortality rates. Without knowing what happened to coagulation in mice who received antibiotics to suggest that this is microbiota changes prevents PE is not correct. 
I believe that you have to discuss that there must be another explanation of your findings. 

Response:

Thanks for your constructive comments. All the antibiotics that we used in the study have such a side effect as thrombocytopenia. To eliminate the influence of antibiotics, we used “microbiota-deficient mice” as receptors in all groups, and transplanted the fresh bacterial suspension of mice treated with vancomycin(donors). In order to remove the impact of residual antibiotics in the bacterial suspension from donors, the feces were washed with PBS at least three times, in Figure 5. We added this part in the Result.

“Before FMT, recipient mice were treated for five consecutive days with 200μl of an antibiotic cocktail by oral gavage to remove their own flora as “receptors” mice. The antibiotic cocktail contained ampicillin (200 mg/kg), vancomycin (100 mg/kg), metronidazole (200 mg/kg) and neomycin (200 mg/kg). The feces of the mice in the NC group and the vancomycin-treated group were collected as donors, and the fecal microorganisms were transplanted into the “microbiota-deficient mice” (receptors). Thereafter, recipient mice were given 200μl of the fresh microbiota suspension by oral gavage three times a week for two weeks.” (Page 15, Line 345 and 352).

Minor remarks in Introduction

1.Please, change the sentence to be as

“Post-thrombotic syndrome can lead to chronic lower leg swelling, which can lead to trophic disorders  including venous ulcers”

Response: Thank you for your advice. We have corrected it (Page 2, Line 58 and 60).

2.In fact, Virchow did not propose a triad. It was given his name later. Please, correct this statement and use for this a more relevant reference.

Response: Thank you for your advice. It’s our mistake. We have revised these statements as following: “The development of clinical thrombosis could be attributed to a combination of vessel wall damage, altered blood flow and abnormal composition of the blood” (reference 12 and 13) (Page 3, Line 69-71).

Round 2

Reviewer 1 Report

Authors addressed comments adequately

Author Response

Dear reviewer,

We appreciate your valuable suggestions on our manuscripts entitled “Early administration of vancomycin inhibits pulmonary embolism by remodeling gut microbiota (Manuscript ID jpm-2203078)". According to your comments, we have carefully revised it.  And we learned a lot from it! We believe that the quality of this revised version has been further improved.

Best regards

Sincerely yours

Reviewer 2 Report

 The Authors did not exclude the possibility that the reduced mortality after thrombin injection was associated with different anticoagulation profile in the animals (perhaps caused by antibiotics), while the observed microbiota differences were just an epiphenomenon. Thus, the conclusion of this study is not supported by the results.

The other problem of this study is that this animal model of PE has nothing to do with actual PE in humans. This animal model is rather a pharmacologically evoked thrombosis in the pulmonary arteries. It is not an embolism. Therefore, even if in this study the outcomes were indeed caused by different microbiota (which is highly questionable), this wouldn't have any impact on the management of PE in humans, where the pathophysiology is completely different

Author Response

Dear reviewer,

We appreciate your valuable suggestions on our manuscripts entitled “Early administration of vancomycin inhibits pulmonary embolism by remodeling gut microbiota (Manuscript ID jpm-2203078)". According to your comments, we have carefully revised it. And we learned a lot from it! We believe that the quality of this revised version has been further improved. The point-to-point responses are as following:

Reviewer(s)' Comments to Author:

The Authors did not exclude the possibility that the reduced mortality after thrombin injection was associated with different anticoagulation profile in the animals (perhaps caused by antibiotics), while the observed microbiota differences were just an epiphenomenon. Thus, the conclusion of this study is not supported by the results.

Response:

Thank you very much for your constructive comments. In fact, we also considered the anticoagulation profile caused by antibiotic in mice. So, in the later experiment of using Fecal microbiota transplantation (FMT) to verify the effect of microbiota, we used "microbiota deficient mice" as the receptor in all groups and transplanted the fresh bacterial suspension of mice (donors) treated with vancomycin. The fresh bacterial suspension was washed with PBS at least three times, to eliminate the effect of residual antibiotics in the bacterial suspension from the donor. The results of FMT proved that intestinal microflora did play a role in the development of PE, and at the same time, to a certain extent, the effect of antibiotic treatment alone on PE was excluded. The FMT method we adopted is a classic method to study the effect of intestinal flora on body function, which has been reported in many studies. (Reference 28, 29)  

“Before FMT, recipient mice were treated for five consecutive days with 200μl of an antibiotic cocktail by oral gavage to remove their own flora as “receptors” mice. The antibiotic cocktail contained ampicillin (200 mg/kg), vancomycin (100 mg/kg), metronidazole (200 mg/kg) and neomycin (200 mg/kg). The feces of the mice in the NC group and the vancomycin-treated group were collected as donors, and the fecal microorganisms were transplanted into the “microbiota-deficient mice” (receptors). Thereafter, recipient mice were given 200μl of the fresh microbiota suspension by oral gavage three times a week for two weeks.” (Page 15, Line 345 and 352).

The other problem of this study is that this animal model of PE has nothing to do with actual PE in humans. This animal model is rather a pharmacologically evoked thrombosis in the pulmonary arteries. It is not an embolism. Therefore, even if in this study the outcomes were indeed caused by different microbiota (which is highly questionable), this wouldn't have any impact on the management of PE in humans, where the pathophysiology is completely different.

Response:

Thank you very much for your valuable suggestions on our manuscript. Pulmonary embolism (PE) is a clinical pathophysiological syndrome caused by the blockage of pulmonary artery trunk or branch by endogenous or exogenous emboli. The clinical research of PE is mainly retrospective analysis, and the basic research on pathophysiology and treatment of PE is mainly based on animal experiments. The PE model we used in this paper is a classic model of mice (reference 24, 25). Due to the blood volume of mice is low, and the jugular and femoral veins are small, it is difficult to prepare and inject thrombus in vitro. It is suitable for the formation of thrombus in situ after drug injection. The PE model can be used for molecular biological experimental research of large-scale samples. Although there may be differences between this model and human pathophysiological process, it also suggests the key role of microbiota in the occurrence and development of PE, and also provides some potential value for the application of more large animal models closer to clinical research. We have further revised the PE model in detail in methodology of the manuscript (Page 4-5, Line 119-128) (reference 24, 25).

“Pulmonary Embolism Mouse Model. Mice were anesthetized with isoflurane. Then, the left side of the internal jugular vein was completely exposed by cutting the skin of the neck, and 15 IU / 30 g thrombin was injected into the jugular vein. Then the mortality rate was recorded in all groups within 30 minutes of the injection (the time of respiratory arrest lasted for at least 2 minutes). Dead mice caused by massive bleeding events were discarded. The left lung lobe was then removed, fixed in 10% formalin and embedded in paraffin. Sections were stained with hematoxylin–eosin. Under the microscope, three fields (X10 objective, X10 ocular) were chosen at random in each section (one section per mouse). The thrombus area of lung tissue was evaluated in each field which was performed blind to groups [24, 25].”

Reviewer 3 Report

Dear Authors! Thank you for addressing my remarks. The paper was improved.

Author Response

(The authors gave the same response as above.)
